# Preparation and Experimental Study of Phase Change Materials for Asphalt Pavement

**DOI:** 10.3390/ma16176002

**Published:** 2023-08-31

**Authors:** Zhuqiang Huang, Jianguo Wei, Qilin Fu, Yuming Zhou, Ming Lei, Zhilong Pan, Xiangchao Zhang

**Affiliations:** 1School of Traffic and Transportation Engineering, Changsha University of Science and Technology, Changsha 410114, China; huangzhuqiang@ccsu.edu.cn (Z.H.); fuqilin100@163.com (Q.F.); zym_2015@csust.edu.cn (Y.Z.); 2College of Civil Engineering, Changsha University, Changsha 410022, China; lm2656717@163.com (M.L.); csxypzl@163.com (Z.P.); xczhang@ccsu.edu.cn (X.Z.); 3Innovation Center for Environmental Ecological & Green Building Materials of CCSU, Changsha University, Changsha 410022, China

**Keywords:** asphalt mixtures, ceramsite (CS), myristic acid (MA), phase change materials (PCMs), paraffin wax (PW)

## Abstract

This study aimed to address the issue of high-temperature challenges in asphalt pavement by developing two types of phase change materials (PCMs) for temperature control. Encapsulated paraffin wax particles (EPWP) and encapsulated myristic acid particles (EMAP) were synthesized using acid-etched ceramsite (AECS) as the carrier, paraffin wax (PW) or myristic acid (MA) as the core material, and a combination of epoxy resin and cement as the encapsulation material. The investigation encompassed leakage tests on PCMs; rutting plate rolling forming tests; SEM, FTIR, XRD, and TG-DSC microscopic tests; as well as heat storage and release tests and temperature control assessments using a light heating device. The study revealed the following key findings. Both types of PCMs exhibited no PCM leakage even under high temperatures and demonstrated low crushing ratios during rut-forming tests. Microscopic evaluations confirmed the chemical stability and phase compatibility of the constituents within the two types of PCMs. Notably, the phase change enthalpies of EPWP and EMAP were relatively high, measuring 133.31 J/g and 138.52 J/g, respectively. The utilization of AECS as the carrier for PCMs led to a substantial 4.61-fold increase in the adsorption rate. Moreover, the PCMs showcased minimal mass loss at 180 °C, rendering them suitable for asphalt pavement applications. The heat storage and release experiments further underscored the PCMs’ capacity to regulate ambient temperatures through heat absorption and release. When subjected to light heating, the maximum temperatures of the two types of phase change Marshall specimens were notably lower by 6.6 °C and 4.8 °C, respectively, compared to standard Marshall specimens. Based on comprehensive testing, EPWP displayed enhanced adaptability and demonstrated substantial potential for practical implementation in asphalt pavements.

## 1. Introduction

Asphalt mixtures, classified as viscoelastic-plastic materials [1,2], are highly sensitive to temperature variations. During the elevated temperatures of summer, asphalt pavements have a tendency to absorb heat, leading to reduced pavement stiffness, rutting, pushing, oil flooding, and other detrimental conditions. These factors significantly curtail the lifespan of asphalt pavements while simultaneously escalating maintenance expenditures [3]. Moreover, the prevalence of high-temperature asphalt pavements exacerbates the urban heat island effect and accelerates the release of noxious constituents present in asphalt. To counteract the detrimental consequences of elevated temperatures on asphalt pavements, several strategies have been proposed:

Enhancing the high-temperature stability of the asphalt pavement itself is achieved through approaches such as incorporating modified asphalt [4], introducing fibers [5,6], and optimizing grading [7]. Mitigating heat transfer into the pavement by applying coatings for heat reflection [8] augments pavement thermal resistance [9,10]. The heat absorption and storage capabilities of asphalt pavements can be enhanced by integrating phase change materials (PCMs), thereby creating phase change asphalt pavements. This active temperature control technique empowers PCMs to autonomously absorb heat and regulate asphalt pavement temperatures. A body of research has explored the application of PCMs within the realm of asphalt pavement. For instance, Montoya, M.A. [11] harnessed the energy storage potential of phase change microcapsules to mitigate the detrimental impacts of harsh environmental conditions on asphalt pavements. Deng, Y. [12] employed SiO_2_ as a carrier to adsorb polyethylene glycol 4000, effectively introducing a PCM. This alteration reduced rutting depth by 10% over one month during the summer by incorporating 10% of this PCM in place of aggregate. Similarly, Jin, J. [13] employed diatomaceous earth as a carrier to adsorb stearic acid and palmitic acid and achieved 8.11 °C and 6.36 °C reductions, respectively, in the temperature of the uppermost asphalt mixture layer upon replacing 0.075 mm fine aggregate and filler. Cakti, K. [14] utilized n-tetradecane as the PCM by incorporating hydroxy ethyl acrylate and hydroxy ethyl methacrylate as copolymerized monomers to create highly heat- and permeation-resistant microcapsules. Athukorallage, B. [15] introduced PCM to asphalt pavements and analyzed the heat transfer process using the volume average energy equation. Anupam, B.R. [16] developed core–shell encapsulated PCM doped with OM-35 and OM 42, resulting in pavement temperature reductions of 3.05 °C and 4.36 °C, respectively. The microcapsules exhibited thermal stability up to 400 °C with a 90% survival rate for milling at 140 °C. Kerahroudi, F.S. [17] investigated the performance of four PCMs doped into diatomaceous earth and expanded perlite. Their study concluded that polyethylene glycol improved the low-temperature performance of asphalt mastic but had negligible effects on medium- and high-temperature behavior. Yuan, J. [18] examined vitrified PCMs and found PCM-43 to be more effective than PCM-48 in regulating asphalt pavement temperatures. Kakar, M.R. [19] introduced tetrade-cane as a PCM to asphalt, leading to increased permeability, lowered softening temperature, and modified complex modulus, significantly influencing the asphalt’s rheological properties. Ryms, M. [20] employed porous-aggregate-adsorbed paraffin wax (PW) as a PCM and studied its thermodynamic parameters. Experimental results indicated a reduction of 5 °C in the surface temperature of PCM-doped asphalt pavements. Finally, Betancourt-Jimenez, D. [21] proposed suitable phase change microcapsules tailored for asphalt pavements, addressing the phase change stability of these microcapsules.

Various PCMs can be categorized into three distinct classes based on their chemical composition. The first category comprises organic PCMs, such as alkanes, fatty acids, and alcohols. The second category encompasses inorganic salt-based PCMs, exemplified by compounds such as Na_2_SO_4_-10H_2_O [22] and CaCl_2_-6H_2_O [23]. Additionally, there exists a class of composite PCMs combining organic and inorganic components, such as mixed combinations of sodium acetate, stearic acid, and octadecanol [24]. Among these, organic PCMs are commonly favored due to their notable enthalpy of phase change, absence of supercooling, thermal stability, and cost-effectiveness. Paraffin and myristic acid (MA) exemplify these organic materials, demonstrating the general advantages of this category. The selection of these materials for this study was driven by their phase change enthalpy and compatibility with the softening point temperature of asphalt.

The carrier materials employed can take the form of porous substances, including porous ceramic grains [25], expanded perlite [26], and expanded vermiculite [27]. Porous ceramic grains are particularly prevalent due to their inherent qualities, such as porosity, renewability, and controllable particle size. Prior studies involving ceramic grains as adsorption carriers have predominantly used pretreated ceramic grains for adsorption, as demonstrated by Li, H. [28], Wang, X. [29], He, H.T. [30], Xie, P.L. [31], Wang, R. [32], etc. However, the observed adsorption rates in these cases remained below 60%, signifying a relatively modest level of adsorption. In contrast, the current investigation treated ceramic grains with a dilute HF solution under negative pressure. This treatment induced the erosion of internal pore walls, thinning of pore walls, opening of previously closed pores, and consequently, a substantial increase in the adsorption rate. Notably, the utilization of acid-etched ceramic grains as carriers for PCMs has not been previously reported.

The encapsulation materials commonly employed are often organic in nature, including substances such as epoxy resin [33] and melamine formaldehyde resin [34]. However, the encapsulation material adopted in this study represents a hybrid composition of organic and inorganic components, utilizing an adhesive blend of epoxy resin and cement. To the best of our knowledge, this particular combination has not yet been reported.

Building upon the aforementioned studies, an intriguing research avenue emerges: the application of acid-etched ceramic grains as carriers to adsorb paraffin and MA under negative pressure, followed by double encapsulation using a hybrid adhesive of epoxy resin and cement to create PCMs. This novel approach has yet to be explored from the perspectives of both carrier material and encapsulation substance. Guided by this concept, the objective of this paper is to introduce a fresh perspective on the potential utilization of PCMs for temperature control in asphalt pavements.

## 2. Experimental

### 2.1. Materials

In this study, ceramsite (CS) was manufactured at Anhui Yatao Ceramsite Factory, featuring a particle size ranging from 9.5 to 13.2 mm. Through comprehensive testing, the bulk density of CS was determined to be 0.189 g/cm^3^, while the cylinder compressive strength measured 0.26 MPa. The utilized PW originated from Shanghai Joule Wax Co. Ltd.(Shanghai, China), possessing a phase change temperature of 55 °C. Similarly, MA was sourced from Nanjing Done Co., Ltd. (Nanjing, China), exhibiting a phase change temperature of approximately 55 °C. The epoxy resin and curing agent were procured from Sinopec Baling Petrochemical Co., Ltd. (Yueyang, China). The epoxy resin, classified as type E51, exhibited an epoxy equivalent of 185.6 g/mol and a viscosity of 13,480 MPa·s at 25 °C. Correspondingly, the curing agent, denoted as T31, possessed an amine value of 550 Mg/KOH/g and a viscosity of 1253 MPa·s at 25 °C. The cement selected for this study was a PC32.5-type product from South Cement Company Limited (Changsha, China).

### 2.2. Preparation of PCMs

First, the CS was subjected to sieving procedures, and CS particles measuring 9.5 mm to 13.2 mm were isolated for subsequent testing, as illustrated in Figure 1a.

The process of vacuum acid etching was executed utilizing custom-designed equipment developed by the research team, as depicted in Figure 1b. This equipment comprised a reaction container, a piston, and a clump weight situated atop the piston, which could be elevated using a string mechanism. The CS, along with the HF solution, was introduced into this apparatus. The piston, assisted by the clump weight, immersed the CS into the HF solution. This arrangement was subsequently introduced into a vacuum chamber for the acid etching process. The acid solution was prepared by diluting a 40 wt.% analytically pure HF solution with deionized water to attain a concentration of 12 wt.%. The quantity of acid solution employed was approximately five times the mass of CS. The acid etching process was carried out for a duration of 6 min, with an evacuating pressure maintained at 2 kPa ± 1 kPa. Following vacuum acid etching, the acid-etched ceramsite (AECS) was subjected to drying for subsequent use.

The preparation of PCMs was executed in accordance with the method elucidated in the author’s patented invention, as illustrated in Figure 1c–f. The PCM was liquefied within an oven, and the vacuum AECS was introduced into the liquid PCM. The amalgamated mixture was then placed within an evacuating apparatus and exposed to a vacuum environment. Once the AECS had adsorbed the PCM completely, the vacuum process was terminated, and the mixture was allowed to cool. Following the cooling phase, the AECS impregnated with the PCM was carefully extracted.

For the next stage, the epoxy resin adhesive E51, curing agent T31, and cement were blended according to the specifications outlined in the patent and thoroughly mixed. Phase change ceramsite (PCC) was introduced into the mixed adhesive and thoroughly agitated to ensure comprehensive coating. Subsequently, the PCC enveloped in the adhesive mixture was introduced into a granulator containing cement powder, resulting in the formation of spherical particles. The cement-coated PCC was extracted through screening and subjected to a drying process. This coating procedure was repeated once more to culminate in the successful preparation of PCMs.

A comparison between acid-treated and untreated ceramic grains yielded noteworthy findings: a pre-acid-etching adsorption rate of 43.33% for untreated grains, and a remarkable post-acid-etching adsorption rate of 199.56%, signifying a substantial 4.61-fold increase following the acid etching procedure. The adsorption rate of untreated ceramic grains aligned with the rates documented in prior literature and exhibited a corresponding 4.61-fold augmentation post-acid etching. This phenomenon can be attributed to the presence of open and closed pores within the grains. While untreated grains solely utilize open pores for adsorption, closed pores remain unutilized, resulting in wasted volume incapable of energy storage. Moreover, the presence of closed air in these pores contributes to reduced thermal conductivity (as explored in subsequent research), underscoring the significance of harnessing closed pores for improved performance.

Density assessments were conducted on encapsulated paraffin wax particles (EPWP) and encapsulated myristic acid particles (EMAP), yielding measured densities of 1450 kg/m^3^ and 1410 kg/m^3^, respectively.

### 2.3. Leakage Test

A series of heat treatment steps were executed for EPWP, EMAP, and encapsulated ceramsites (ECS). Approximately 30 g of each particle type was placed in Petri dishes lined with qualitative filter papers and subjected to an oven environment. Initially, the oven was set at 80 °C for 2 h, followed by a temperature adjustment to 180 °C for an additional 2 h. Lastly, the temperature was reverted to 80 °C for a prolonged 96 h duration, culminating in a total baking time of 100 h. The mass of the particles was measured every 2 h, and any changes in mass and potential leakage were meticulously documented.

### 2.4. Rut Forming/Crushing Test

To create an AC-20 asphalt mixture with a 4.7% oil/stone ratio, the 9.5 mm–13.2 mm stones were substituted with equivalent volumes of EPWP and EMAP. The aggregate ratios employed are outlined in Table 1. Rutting slabs containing phase change particles were formulated in accordance with the procedures stipulated in AASHTO TP63-07. Subsequently, trichloroethylene was employed to dissolve the asphalt within the rutting slabs, allowing for the separation and weighing of the PCMs, both broken and intact. The assessment of the PCM crushing ratio provided insight into whether the PCM-infused asphalt met the requisite standards for normal pavement paving.

### 2.5. Microscopic Characteristics

The PCMs underwent 20 thermal cycles and were dissected to reveal small flaps, with a portion reserved for SEM testing. Another segment of the small flaps was processed using a sample-making crusher, generating powdered samples for FTIR, XRD, and TG-DSC examinations. FTIR and XRD analyses aimed to identify the potential emergence of novel functional groups or substances.

SEM Testing: Gold spraying was employed to treat the specimens, followed by observation using a Japanese Hitachi SU-1510 scanning electron microscope (SEM) (Hitachi, Co., Ltd., Tokyo, Japan). Six interfaces were meticulously selected for observation: the inner regions of CS and AECS, the interfaces of EPWP and EMAP, the interface between AECS and the encapsulation material, and the encapsulation material itself.

FTIR Testing: The phase change particle powder was compressed into specimens using KBr (Tianjin Kermel Co., Ltd., Tianjin, China) and a tablet press. Measurements were conducted using a Nicolet iS5 FTIR spectrometer (Thermo Fisher Scientific Inc., Waltham, MA, USA). The scanning range spanned from 4000 to 400 cm^−1^, and the scanning frequency encompassed 16 repetitions for each analysis.

XRD Testing: Phase change particle powder was subjected to XRD testing utilizing a Shimadzu XRD-6100 instrument (Shimadzu Co., Ltd., Kyoto, Japan). The parameters included k = 0.154 nm, a voltage of 40 kV, and a current of 200 mA. The scanning range was set from 5 to 80°, with a scanning speed of 5°/min.

### 2.6. Thermal Performance

TG Analysis: Thermogravimetric analysis was carried out using a NETZSCH STA 449 F3 (NETZSCH-Gerätebau GmbH, Selb, Germany) simultaneous thermal analyzer under nitrogen protection. The analysis encompassed a temperature range from room temperature to 900 °C.

DSC Testing: A TA DSC 250 instrument (TA Instrument Co., Ltd., New Castle, DE, USA) was employed for DSC analysis. The test temperature spanned from room temperature to 90 °C, with a heating rate of 5 °C/min and a duration of 5 s under argon gas protection.

### 2.7. Phase Change Particle Heat Storage/Release Performance Test

Thermostatic Water Tank Testing: The experimental setup included two CT-25 thermostatic water tanks, as depicted in Section 3.7. Specifically, 100 g of PW, MA, CS, EPWP, and EMAP were placed into separate beakers, each equipped with an electronic thermometer probe positioned at the center of the specimen. The beakers were covered with foam lids and sealed using plastic bags. The temperature at the center of each specimen was recorded. Two thermostatic water baths, set at 30 °C and 80 °C, were prepared for temperature rise and drop testing. For the temperature rise examination, the specimens were immersed in a 30 °C water bath until their center temperatures reached 30 °C. Subsequently, the specimens were swiftly transferred to an 80 °C water bath, and temperature readings were taken every 30 s until all specimens attained temperature equilibrium. In the temperature drop assessment, the specimens were promptly placed back into the 30 °C constant-temperature water bath after the temperature rise test, and the temperature was recorded every 30 s until the specimens reached temperature equilibrium.

### 2.8. Marshall Specimen Temperature Rise/Drop Test

Marshall specimens (MS) were crafted through compaction following the method outlined inJTG E20 [35]. For the purpose of temperature rise and drop tests, the MS were subjected to a specially designed light heating device, as depicted in Section 3.8. This device was employed to gauge the thermal behavior of the MS. Three variations of MS were prepared: ceramsite Marshall specimens (CSMS), encapsulated paraffin wax particle Marshall specimens (EPWPMS), and encapsulated myristic acid Marshall specimens (EMAPMS). These were generated by substituting aggregates in the range of 9.5 mm to 13.2 mm with CS, EPWP, or EMAP of equivalent volume. Notably, the standard MS underwent 75 compactions on both sides, whereas CSMS, EPWPMS, and EMAPMS received 10, 25, and 25 compactions, respectively, on both sides. Within EPWPMS, EPWP accounted for 12.16% of the mass, EMAP accounted for 11.91% of the mass within EMAPMS, and CS accounted for 3.98% of the mass within CSMS. All AC-20 MSs were trimmed to a height of 64 mm, with slots at the bottom center to accommodate temperature sensor probes. These probes were sealed using asphalt, and the slots were subsequently filled.

Employing a custom-designed light heating device, the specimens were gradually heated from room temperature to 70 °C, followed by a controlled cooling phase from 70 °C back to room temperature. The resulting temperature curves were plotted and analyzed. This light heating device featured two 275 W infrared lamps as heat sources. Placing these lamps mid-perpendicular between two adjacent MS ensured uniform heat distribution on their surfaces. The MS were housed in foam boxes filled with fine-grained expanded perlite particles chosen for their heat-retention properties. Temperature readings were recorded every 2 min, contributing to the creation of temperature rise and drop curves for the four MS.

## 3. Results and Discussion

### 3.1. Leakage Test

During the leakage test, temperatures of 180 °C and 80 °C were chosen to simulate different conditions relevant to PCMs in asphalt pavement materials, including heating, mixing, paving, and working temperatures. The absence or minimal occurrence of PCM leakage was considered a favorable outcome.

The external morphology of the PCMs after 100 h of testing is depicted in Figure 2a. At each weighing interval, the surfaces of the PCMs remained relatively clean, showing no signs of PCM leakage. The qualitative filter papers within the Petri dishes displayed no indication of staining from the PCM. Notably, both EPWP and EMAP exhibited no signs of leakage.

The masses weighed every 2 h were plotted, as shown in Figure 2b. The initial mass of ECS was 30.189 g, which decreased to 29.803 g after 100 h of testing, resulting in a mass loss of 1.28%. For EPWP, the initial mass was 30.298 g, decreasing to 29.902 g after 100 h, corresponding to a mass loss rate of 1.31%. Similarly, the initial mass of EMAP was 30.523 g, and after 100 h, the mass decreased to 30.127 g, yielding a mass loss rate of 1.30%. The mass losses among the three particle types were nearly identical, and all were below 1.31%. As depicted in Figure 2b, the highest mass losses for the three particle types occurred within the initial 4 h. During this period, the mass loss of EPWP constituted 85.86% of the total, EMAP’s mass loss contributed 88.38%, and ECS’s mass loss accounted for 86.10%. This high initial mass loss underlines the impact of elevated temperatures. The loss of mass was predominantly attributed to the encapsulation material. Moreover, the higher temperatures at the outset led to greater mass losses, whereas lower temperatures in the subsequent phases resulted in comparatively lower mass losses. This phenomenon can be interpreted as follows: elevated temperatures triggered evaporation of epoxy resin within the encapsulation material, leading to drying of the PCMs and detachment of cement particles from the surface. These mass loss processes were more pronounced at higher temperatures initially and slowed down as temperatures dropped in the middle and later stages. Given that the mass loss of PCMs was confined to 1.31%, it can be disregarded in the context of asphalt pavement construction.

Combining the observations of PCM behavior and the mass curves, a conclusive assessment can be drawn. Both EPWP and EMAP exhibited superior encapsulation qualities, demonstrating no leakage of PCM. As a result, these materials can be deemed suitable for utilization in asphalt pavement applications.

### 3.2. Rut Forming/Crushing Test

The process of preparing rutting slabs involves subjecting them to high temperature and pressure, simulating conditions akin to those encountered during asphalt paving. The analysis of the PCM crushing ratio is instrumental in assessing their strength and adherence to construction specifications. Lower crushing ratios signify enhanced strength and better alignment with construction demands.

Both EPWP and EMAP rutting slabs were divided into four equal quarters, and one of these quarters was randomly chosen for asphalt dissolution using trichloroethylene. Subsequently, the EPWP and EMAP were extracted from the dissolved quarters, air-dried, and weighed, as visually illustrated in Figure 3. The cumulative mass of the extracted EPWP amounted to 261.24 g, constituting 25.16% of the total EPWP mass. Within this, the mass of crushed EPWP was measured at 15.37 g, corresponding to a crushing ratio of 5.56%. Similarly, the collective mass of the extracted EMAP stood at 247.61 g, accounting for 24.86% of the total EMAP mass. Among these, the mass of crushed EMAP was recorded as 18.15 g, yielding a crushing ratio of 6.83%. While the crushing ratio of EMAP exceeded that of EPWP, both remained below 7%, aligning well with the stipulated asphalt paving requirements.

### 3.3. Scanning Electron Microscopy

As illustrated in Figure 4a, CS exhibits fewer internal pores, with minimal connectivity among them and a predominance of closed pores accompanied by thicker pore walls. Conversely, Figure 4b demonstrates the outcome of acid etching, revealing a central portion of CS characterized by increased large and interconnected pores coupled with thinner pore walls. Consequently, AECSs display a substantially enhanced pore size and heightened capacity for adsorbing PCMs due to the effect of acid etching. These observations find validation in the subsequent outcomes as well. Prior to acid etching, the adsorption rate of CS was measured at 43.33%, which surged to 199.56% following acid etching, resulting in a remarkable 4.61-fold increase. This enhanced adsorption can be attributed to the augmented porosity and pore size consequent to acid etching. Figure 4c,d corroborate these findings by depicting acid-etched ceramsite particles amply adsorbed with PW and MA. Notably, MA clusters together, forming distinct blocks that are clearly demarcated from the AECS shells. Further observations in Figure 4d,e highlight the permeation of the porous AECS shell by the PCM, ensuring excellent contact with the shell. Additionally, the interface between the AECS shell and the encapsulation material is conspicuously evident, with the exterior enveloped and permeated by the encapsulation material. Figure 4f provides a glimpse of the encapsulation material, comprising a blend of epoxy resin and cement. The presence of sizable cement particles within the spherical grains is noticeable. These cement particles are evenly dispersed within the encapsulation material to bolster the hardness and high-temperature resilience of the epoxy resin shell, in line with prior research [36,37,38].

### 3.4. FTIR Analysis

Displayed in Figure 5a, PW exhibits distinct characteristic peaks as follows: one at 2917 cm^−1^ attributed to the -CH3 stretching vibration caused by C-H; another at 2849 cm^−1^ related to the -CH2 stretching vibration induced by C-H [39,40]; the peak at 1468 cm^−1^ linked to the -CH3 asymmetric bending vibration; and one at 720 cm^−1^ assigned to the -CH2 rocking vibration and deformation vibration [41]. Notably, EPWP displays similar characteristic peaks at 2917 cm^−1^, 2849 cm^−1^, 1471 cm^−1^, and 720 cm^−1^, mirroring those of PW. Hence, it can be inferred that no chemical reaction has occurred between PW, CS, or the encapsulation shell, and their interaction remains at a physical mixing level.

Figure 5b shows the distinctive absorption peaks associated with MA. A -CH2 symmetric stretching vibration absorption peak emerges at 2918 cm^−1^ alongside a stretching vibration peak corresponding to C=O in the carboxyl group at 1701 cm^−1^ [42,43]. An asymmetric bending vibration peak of -CH2 occurs at 1465 cm^−1^ [44]. Notably, the in-plane and out-of-plane bending vibration peaks of -OH are observable at 1420 cm^−1^ and 938 cm^−1^, respectively. The peak at 1287 cm^−1^ is attributed to the C-O stretching vibration, while the 721 cm^−1^ peak corresponds to the surface bending vibration of C-H [45]. Similarly, EMAP manifests corresponding peaks at 2919 cm^−1^, 1701 cm^−1^, 1465 cm^−1^, 1287 cm^−1^, 938 cm^−1^, and 721 cm^−1^. The IR spectra of both EMAP and MA are strikingly similar, with no emergence or disappearance of new characteristic peaks. Consequently, it can be ascertained that no new functional groups were generated within the PCMs.

### 3.5. XRD Analysis

As depicted in Figure 6a, PW exhibits more prominent diffractive absorption peaks at 2θ values of 21.39° and 23.77° [46] alongside less intense diffractive absorption peaks at 40.34° and 42.30°. Notably, the characteristic peaks of EPWP appear as a blend of the characteristic peaks of both PW and CS, suggesting the absence of any chemical reaction or the formation of new compounds when PW and CS are combined. EMAP, on the other hand, displays distinctive diffraction absorption peaks at 2θ values of 14.00°, 21.52°, 24.08°, and 40.76° [47,48]. A comparative analysis of the characteristic diffraction peaks of EMAP, MA, and CS reveals that while the positions of these peaks are fundamentally identical, the peak intensities have slightly decreased. This finding implies that no chemical reaction transpired among EMAP, CS, and the encapsulation shell and that no new material has emerged.

Hence, it can be definitively concluded that the internal constituents of both EPWP and EMAP are merely combined without undergoing any chemical reactions, indicating their compatibility.

### 3.6. TG-DSC Analysis

(1)TG analysis

Higher TG decomposition temperatures correspond to improved heat resistance and temperature stability. The TG test outcomes for CS, PW, MA, EPWP, and EMAP are depicted in Figure 7. PW mass declines rapidly between 260 and 380 °C, displaying a maximal TG mass loss of 94.47%. Similarly, MA’s mass experiences a rapid reduction between 200 and 303 °C, with a peak TG mass loss of 99.11%. The variations in quality for EPWP are as follows: from 0 to 155 °C, there is almost no alteration; from 155 °C to 288 °C, the PCM initiates decomposition; and from 288 °C to 379 °C, both the PCM and the encapsulation material undergo complete decomposition. Concerning EMAP, the quality changes are as follows: negligible alteration from 0 to 160 °C, PCM decomposition from 160 °C to 288 °C, and comprehensive decomposition of the PCM and encapsulation material from 288 °C to 390 °C. In both EPWP and EMAP, there is a gradual decline in mass between 288 °C and 357 °C, indicating the decomposition of the encapsulation material. The mass reductions observed in the PCMs are mainly attributed to PCM decomposition, with a secondary contribution from encapsulation material decomposition. EPWP’s maximum TG mass loss is 57.53%, signifying a 36.94% reduction compared to PW. Similarly, EMAP’s peak TG mass loss is 57.50%, demonstrating a reduction of 41.61% compared to MA.

Asphalt mixtures are typically heated, mixed, and paved at approximately 180 °C, a temperature at which EPWP and EMAP experience mass losses. Based on the TG mass losses outlined in Table 2, both EPWP and EMAP exhibit mass losses below 5%, which falls within the acceptable range. Therefore, both varieties of PCMs are suitable for asphalt pavement applications. Additionally, EPWP demonstrates superior heat resistance compared to EMAP.

(2)DSC analysis

Greater ΔHm values indicate higher heat absorption with the same mass of PCMs and better overall performance. The DSC test results for PW, MA, EPWP, and EMAP are depicted in Figure 8. PW exhibits a peak phase change temperature of 55.14 °C and a phase transition enthalpy of 198.92 J/g. For EPWP, the peak phase change temperature is 54.58 °C, and the corresponding phase transition enthalpy is 133.31 J/g. The phase change enthalpy of EPWP accounts for 67.02% of that of PW. MA displays a peak phase change temperature of 55.31 °C along with a phase transition enthalpy of 199.63 J/g. For EMAP, its peak phase change temperature is 55.46 °C, accompanied by a phase transition enthalpy of 138.52 J/g. The phase change enthalpy of EMAP is 69.39% of that of MA. Thus, the phase change enthalpies of both EPWP and EMAP are notably high, satisfying the requirements for energy storage materials in asphalt pavement applications.

### 3.7. Phase Change Particle Energy Storage/Release Test Result Analysis

As depicted in Figure 9b, the temperature rise of MA, CS, and PW occurs more rapidly, while the rise of PCMs is comparatively slower. Before reaching 60 °C, EMAP experiences a quicker temperature rise; after 60 °C, EPWP’s temperature ascends more rapidly. The temperature drop of CS is the most rapid, reaching the lower end of the test temperature quickly. Both the PCMs and pure PCMs exhibit three distinct temperature drop stages.

Stage 1: The rapid temperature drop stage spans from 80 °C to approximately 55 °C. In this stage, the temperature plummets quickly for both the PCMs and pure PCMs, with the latter dropping the fastest and the former dropping at a comparatively slower pace. This difference stems from the fact that after the temperature rise phase, the PCM transitions into a liquid state. Due to the greater contact area of the liquid PCM compared to the point contact areas of encapsulated PCMs, the liquid PCM boasts enhanced thermal conductivity, thereby resulting in a more rapid temperature drop.

Stage 2: The temperature drop stabilizes in this stage, falling from approximately 55 °C to approximately 45 °C. The temperature decrease slows down notably for pure PCMs, and the PCMs also experience a deceleration. However, the stagnation of the temperature drop for PCMs is briefer than that of pure phase change materials. Specifically, the temperature drop stagnation stage for PCMs spans approximately 15 min, while for pure phase change materials, it extends to approximately 100 min. This disparity can be attributed to the fact that as the PCM starts emitting heat, the rate of temperature reduction diminishes, resulting in a stage of temperature drop stabilization. Given that PCMs contain fewer phase change substances in comparison to pure PCMs of equal weight, the phase change enthalpy of pure PCMs is greater. Consequently, pure PCMs absorb more heat and take a longer duration to release it.

Stage 3: This stage encompasses the continuous temperature drop phase, spanning from approximately 45 °C to the lower temperature range of the test. During this phase, both the phase change particles and pure phase change materials experience a rapid temperature decline. Although the PCMs continue releasing heat, the intensity of heat release decreases. As the cumulative heat release increases, the equilibrium becomes unsustainable, resulting in a swift temperature reduction.

Specifically, the temperature drop of EMAP is faster than that of EPWP, and the temperature drop of MA is quicker than that of PW. It is important to note that PW exhibits a phase change peak near 35 °C and experiences another stage of heat release near this temperature, contributing to the slower temperature drop observed in PW.

### 3.8. Marshall Specimen Temperature Rise/Drop Results Analysis

Under equal heat input, lower maximum temperatures indicate better temperature regulation. As depicted in Figure 10b, MS exhibits the most rapid temperature rise, followed by CSMS, EMAPMS, and EPWPMS. MS reaches a peak temperature of 69.6 °C, CSMS of 68.07 °C, EMAPMS of 64.8 °C, and EPWPMS of 63 °C. The maximum temperature of EPWPMS is 6.6 °C lower than that of MS, EMAPMS is 4.8 °C lower than MS, and CSMS is 1.53 °C lower than MS. Consequently, EPWPMS demonstrates the most effective temperature regulation, followed by EMAPMS and CSMS, which also display some degree of temperature regulation.

The underlying factors for this phenomenon can be explained as follows: At elevated temperatures, the PCMs within EPWP and EMAP transition from a solid to a liquid state, absorbing heat within the MS. This process achieves the goal of temperature regulation for the MS. While the phase change enthalpies of EPWP and EMAP are comparable, they are used to replace the aggregate in equal volume. Additionally, EPWP accounts for 12.16% of the mass of EPWPMS, whereas EMAP accounts for 11.91% of the mass of EMAPMS. Since the mass of EPWP is greater than that of EMAP, EPWP contains more PCM. Consequently, EPWPMS exhibits superior temperature regulation compared to EMAPMS. Around the phase change temperature, the temperature curves of EPWPMS and EMAPMS display gradual transitions during both the temperature rise and temperature drop phases. The duration of the temperature rise is approximately 20 min, and the temperature drop phase spans approximately 40 min, consistent with the heat storage and release test outcomes of PCMs. This is where the temperature-regulating capacities of EPWP and EMAP come into play in the context of asphalt mixture temperature control. When preparing MS using the equal volume replacement method, EPWP surpasses EMAP in achieving effective temperature regulation.

## 4. Conclusions

(1) Both EPWP and EMAP can be incorporated into asphalt mixtures as PCMs to mitigate the high temperatures experienced by asphalt mixtures. During the simulation of the asphalt pavement paving process involving heating and rolling, no instances of EPWP or EMAP leakage were observed, and the crushing rate remained below 7%.

(2) SEM, FTIR, and XRD analyses confirmed the structural stability of the internal crystals of EPWP and EMAP. Thermal analysis tests revealed phase change enthalpies of 133.31 J/g for EPWP and 138.52 J/g for EMAP; both exhibited decomposition rates of less than 5% at 180 °C. The phase change enthalpies of EPWP and EMAP exceeded that of EMAP. Their larger phase change enthalpies, along with their minimal decomposition at asphalt pavement construction temperatures, make them well suited for this application.

(3) MS, produced by replacing the 9.5 mm–13.2 mm stone material with equal volumes of EPWP and EMAP, demonstrated a reduction in maximum temperature by 6.6 °C and 4.8 °C, respectively, when compared to standard MS. In terms of heat resistance and MS performance, EPWP exhibited superior effects compared to EMAP. The utilization of EPWP as a replacement for aggregate in asphalt pavement paving, particularly in combination with different sizes of ceramic grains, holds the potential to deliver significant cooling benefits. This approach can effectively mitigate the challenge of high temperatures in asphalt pavement, contributing to reduced urban heat island effects and lower emissions of harmful gases from asphalt. These outcomes would lead to substantial economic and societal advantages.

## Figures and Tables

**Figure 1 materials-16-06002-f001:**
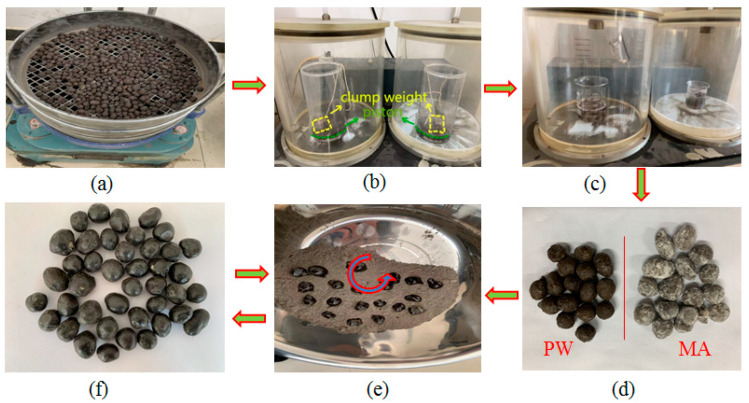
Schematic diagram of the phase change particle preparation process. (**a**) sieving; (**b**) negative pressure acid etching; (**c**) negative pressure wax absorption; (**d**) peeling; (**e**) encapsulation; (**f**) PCMs.

**Figure 2 materials-16-06002-f002:**
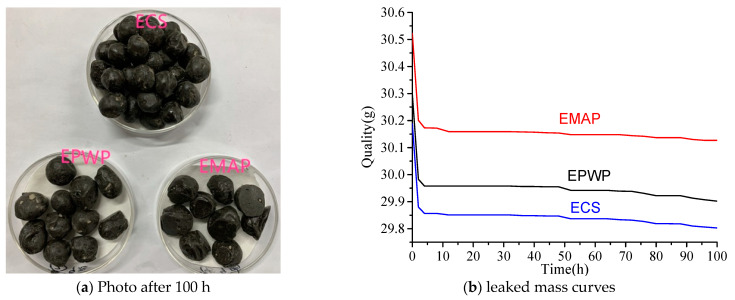
Leakage test.

**Figure 3 materials-16-06002-f003:**
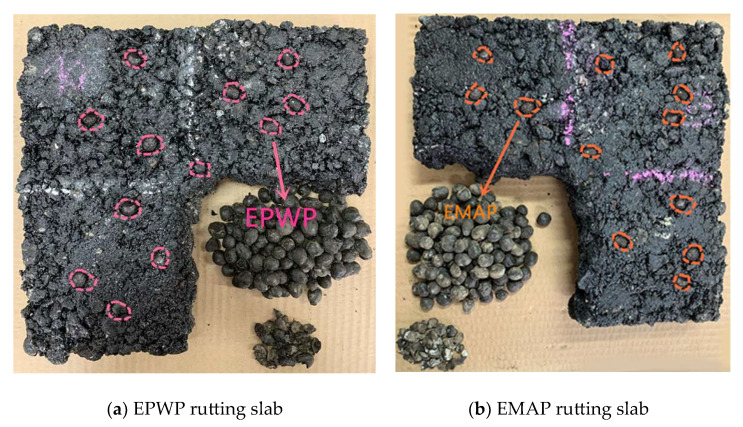
Rutting slabs with one quarter dissolved.

**Figure 4 materials-16-06002-f004:**
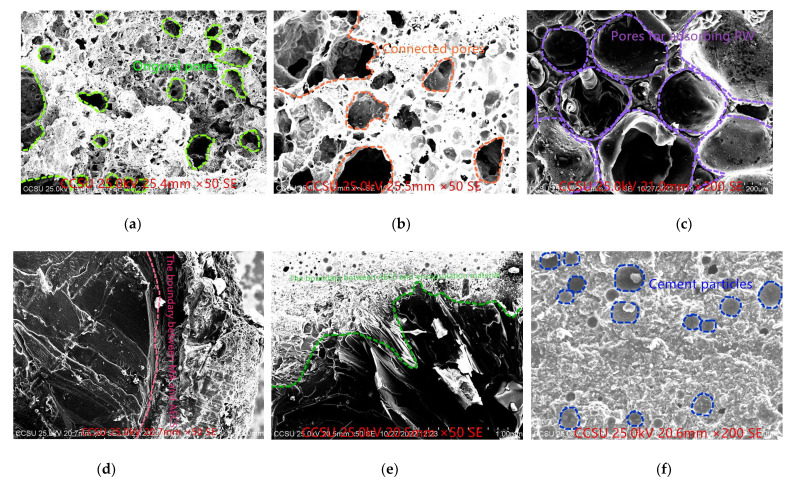
SEM images. (**a**) CS internal view; (**b**) AECS internal view; (**c**) EPWP internal view; (**d**) EMAP internal view; (**e**) AECS shell and encapsulation material interface; (**f**) encapsulation material.

**Figure 5 materials-16-06002-f005:**
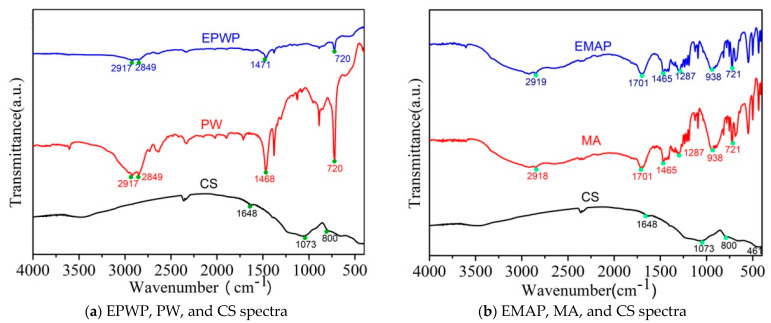
FTIR images.

**Figure 6 materials-16-06002-f006:**
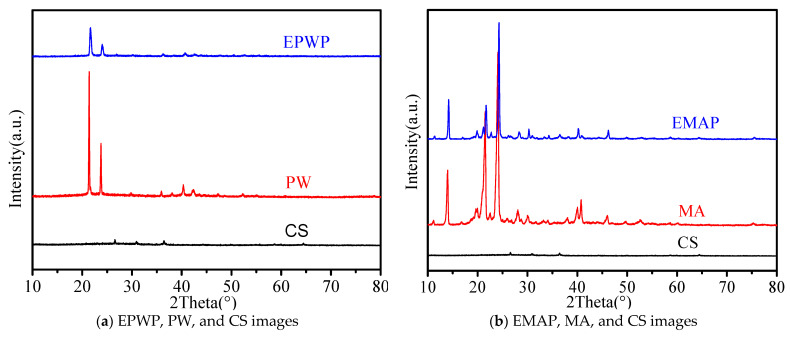
XRD images.

**Figure 7 materials-16-06002-f007:**
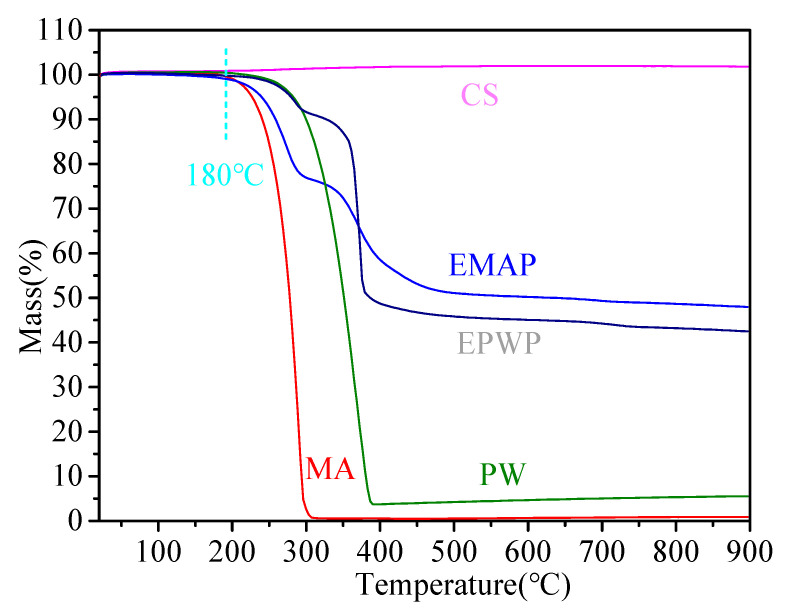
TG images (EPWP, EMAP, MA, PW, and CS mass loss).

**Figure 8 materials-16-06002-f008:**
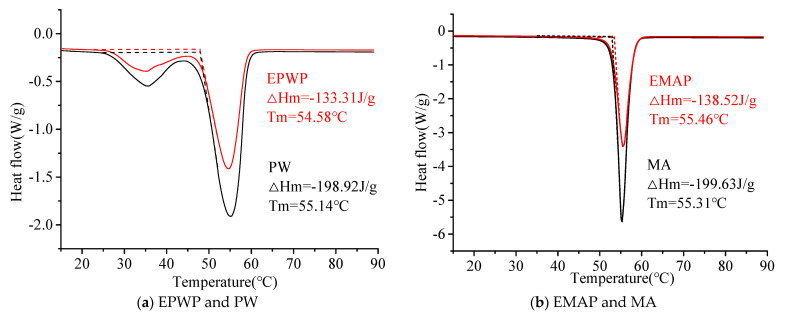
DSC analysis images.

**Figure 9 materials-16-06002-f009:**
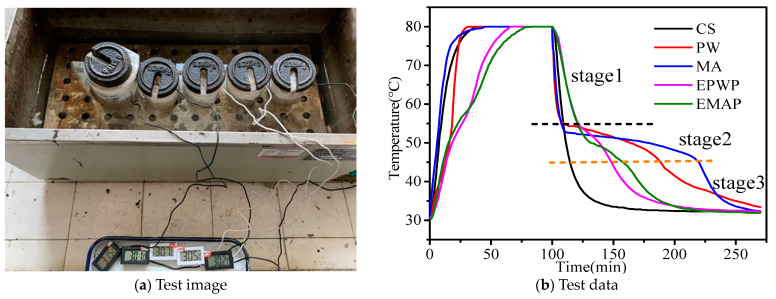
Heat storage and release tests.

**Figure 10 materials-16-06002-f010:**
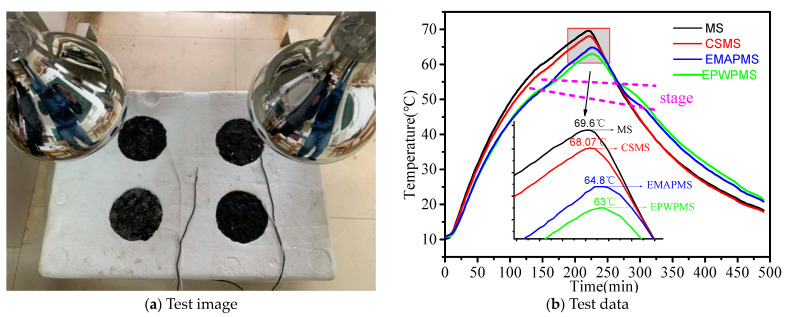
Marshall specimen temperature rise/drop test.

**Table 1 materials-16-06002-t001:** Aggregate ratio table.

Sieve size mm	26.5	19	16	13.2	9.5	4.75	2.36	1.18	0.6	0.3	0.15	mineral powder
Passing rate %	100	96.73	85.48	73.37	56.4	37.04	31.45	25.47	17.49	11.5	8.25	4.33

**Table 2 materials-16-06002-t002:** TG mass losses.

No.	Samples	T-0 (°C)	T-0.05 (°C)	T-0.1 (°C)	T-0.2 (°C)	T-0.5 (°C)	Total Mass Loss (%)
1	CS	-	-	-	-	-	−1.83
2	PW	219.22	284.33	299.59	321.78	350.7	94.47
3	MA	183.64	229.45	238.71	256.3	276.84	99.11
4	EPWP	161.08	280.44	328.93	362.29	389.95	57.53
5	EMAP	110.55	242.45	259.89	284.19	631.42	52.05

## Data Availability

Data are contained within the article.

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
