# Peer review of "Preparation and Experimental Study of Phase Change Materials for Asphalt Pavement"

_materials, 2023, doi:10.3390/ma16176002_

Round 1

Reviewer 1 Report

Review of Manuscript Number: materials-2553560

Full Title:  Preparation and Performance Exploration of Phase Change Materials for Asphalt Pavement

Reviewer notes:

1.     List the keywords in alphabetic order.

2.     Lines 67-38: Please add more discussion about “In contrast, replacing aggregates with PCMs offers huge potential for temperature regulation considering their large proportions in asphalt mixtures”.

3.     Define EPWP, EMAP, HF, PW and MA, lines 70-71.

4.     Define CS in line 81.

5.     The introduction needs to be more emphasized on the research work with a detailed explanation of the whole process considering past, present and future scope. How does the present study give more accurate results than previous studies? It needs to be strengthened in terms of recent research in this area with possible research gaps. It is strongly recommended to add recent literature. Research gaps should be highlighted more clearly and future applications of this study should be added

6.     The study needs to explicitly argue the gaps in the literature and how the current review study will fill those gaps. This is an essential part of this review study which has not been addressed.

7.     The authors need to provide meaningful discussions in their manuscript's Results and Discussion section. The authors need to compare their findings with those of others to add more technical merit to their study.

8.     How many samples were used to obtain reliable statistics of the experimental results?

9.     There are a number of minor issues related to word choice, and acronyms.

 .

Reviewer 2 Report

This paper presents a research study investigated the preparation process and overall performance of phase change materials intended for asphalt pavements. Authors provide analysis focused on two materials: encapsulated paraffin wax and encapsulated myristic acid with the use of acid-etched ceramsite as the PCM carrier. Relevant performance, crushing rate, strength, thermal cycles and thermal analysis tests have also been performed and analysed.

Although presented article may be interesting to the readership of this journal, the paper may only be considered for publication after the following concerns have been addressed successfully in a major revision:

In general:
1) Introduction of the "nomenclature" part is recommended because there are some abbreviations and markings that should be defined at the beginning of the paper. Including this section would make the article much easier for readers less familiar with specific abbreviations.
2) The whole text in my opinion should be readed once again carefully by a english native speaker and "polished" due to some stylistic mistakes. These are not big problems, but some statements could be better worded. There shuold be also made some changes in titles, like for example "Experimental study" or "Experiments" instead of "Experimental".
3) The Introduction needs some attention (please see point 4), "Results and discussion" part and need some upgrade (please see point 5), and finally "Conclusions" paragraph should be presented in better form (please see point 6).

Regarding to the "Introduction" part:
4) In my opinion this part needs some improvement. The Introduction should introduce the Readers to the subject and problems related to the research, and show current state of art. Now, in my opinion, this condition is not entirely fulfilled. Why is it so important to search for new forms of surface temperature reduction? Why to use carrier for PCMs? What are the other ways to achieve similar results? What are the problems and advantages of the new solution? Please make it visible what was the purpose of the research and what made it innovative or important. I know that question is obvious, but you need to introduce the Readers to the subject and prove or set up the thesis. In my opinion these things have not even been outlined properly here. What is more, similar research were performed and described in two papers I know: "The use of lightweight aggregate saturated with PCM as a temperature stabilizing material for road surfaces" (https://doi.org/10.1016/j.applthermaleng.2015.02.036) and "Thermal stabilization and permanent deformation resistance of LWA/PCM-modified asphalt road surfaces" (https://doi.org/10.1016/j.conbuildmat.2017.03.050). Please compare your research with those and show, what progress was made and what is an innovation in your paper.

Regarding to "Results and discussion" part:
5) Authors show many results, but as far as I understand there is no direct comparison with standard asphalt pavement samples. Please make come comparison including some percentage comparison. How better is the solution you purpose comparing with traditional pavement composition? I also do not found any calculus of measurement uncertainties. Please also compare (if able) your results with papers mentioned in point 4.

Regarding to the "Conclusions" part:
6) Also this part needs some improvement. In my opinion whole this section is too general and too descriptive, and does not relate essentially to the results. Good idea would be pointed out what have been done, then present the conclusions, results, improvements etc. What should be done next? What is the significance of this manuscript in the context of the results obtained? Unfortunately, Authors do not answer many of these questions and this is a great place to do so.

Minor errors:
7) Units, such as cm-1, do not have superscripts correctly marked. Same situation with subscripts for -CH2, -CH3 etc.
8) EPWP, PW, and CS spectra from Figure 5(a) could be placed on one graph with EMAP, MA, and CS spectra from Figure 5(b) because the scales on both graphs are exactly the same (or not?) Please check it. Same situation for Fig. 6 and 7.

Minor editing of English language required.

Reviewer 3 Report

Dear Authors,

Thank you for your work. Below, I'm posting my comments:

1) In my opinion, the review of the literature carried out for the asphalt binder is not very international and does not refer to world and European research - it should be expanded and include, among others, information on the climate in which the designed and modified asphalt pavements are or will be used/exploited.

2) Line 39: The Authors write about the modification of asphalt mixtures with fibers - please add what fibers you mean and then give the citation.

3) The title of the article is: "Preparation and Performance Exploration of Phase Change Materials for Asphalt Pavement", but the literature review and the purpose of the work did not specify what phases are even approximately? Amorphous phases are thermodynamically variable and metastable and please consider this information in the analysis of the literature.

4) In the SEM test, open and closed pores were written - these are assumptions, especially since they are qualitative tests, a better test to determine the porosity and types of pores (open and closed) and the porous skeleton would be a computer tomography analysis (qualitative and quantitative test) . Please consider this in future studies describing the porous framework.

5)The Authors did not clearly show which phases we are dealing with in the existing modification and which phases undergo qualitative or quantitative changes. This is important from the assumption of the title. The tests are described in a correct and detailed manner, but the microstructure and structure of the samples/test material is described in general terms.

6) When analyzing asphalt pavements and changes in the structure of the material in terms of temperature changes/increase, refer to thermodynamic databases and provide information on the phases present in the base material/reference material, and then speculate and analyze what changes could have occurred under the influence of the applied factors. I miss that information in this article. The Authors described the research well, but chaotically referred to the subject of the work, i.e. phase analysis.

The article is well written and documented in terms of presenting the process, plan and course of research. The review of the literature and the analysis of phase structures, which is mentioned in the title of the article and what is missing, should be improved.

Phase changes (what phases?) occur mainly under the influence of pressure and temperature, as natural factors (especially ambient temperature for a specific climatic zone in which the asphalt surface is used) and information on the thermodynamic stability of individual phases (what?)

Thank you and Best Regards,

R.

Round 2

Reviewer 1 Report

The authors have addressed all the comments and the manuscript can be accepted.

Reviewer 2 Report

In General:
This paper presents definitely better than before. My previous comments have been correctly addressed in this Review. This paper in its current form may in my opinion be approved for publication.

Minor errors:
- There are still some superscripts that are not correctly marked like SiO2 in line 55 for example.

Reviewer 3 Report

Dear Author,

Thank you for your work and corrections to this paper.

Regards